# Insights into Hunter syndrome from the structure of iduronate-2-sulfatase

Mykhaylo Demydchuk[1,*], Chris H. Hill[1,*,†], Aiwu Zhou[2], Gábor Bunkóczi[1], Penelope E. Stein[3,†], Denis Marchesan[3], Janet E. Deane[1] & Randy J. Read[1]

Hunter syndrome is a rare but devastating childhood disease caused by mutations in the *IDS* gene encoding iduronate-2-sulfatase, a crucial enzyme in the lysosomal degradation pathway of dermatan sulfate and heparan sulfate. These complex glycosaminoglycans have important roles in cell adhesion, growth, proliferation and repair, and their degradation and recycling in the lysosome is essential for cellular maintenance. A variety of disease-causing mutations have been identified throughout the *IDS* gene. However, understanding the molecular basis of the disease has been impaired by the lack of structural data. Here, we present the crystal structure of human IDS with a covalently bound sulfate ion in the active site. This structure provides essential insight into multiple mechanisms by which pathogenic mutations interfere with enzyme function, and a compelling explanation for severe Hunter syndrome phenotypes. Understanding the structural consequences of disease-associated mutations will facilitate the identification of patients that may benefit from specific tailored therapies.

[1] Cambridge Institute for Medical Research, Wellcome Trust/MRC Building, Cambridge Biomedical Campus, Hills Road, Cambridge CB2 0XY, UK. [2] Key Laboratory of Cell Differentiation and Apoptosis of Ministry of Education of China, Shanghai Jiaotong University School of Medicine, Shanghai 200025, China. [3] Department of Medicine, University of Cambridge, Cambridge Biomedical Campus, Hills Road, Cambridge CB2 0SP, UK. * These authors contributed equally to this work. † Present address: Haematological Medicine, King's College Hospital NHS Foundation Trust, Denmark Hill, Camberwell, London SE5 5RS, UK (P.E.S); MRC Laboratory of Molecular Biology, Francis Crick Avenue, Cambridge Biomedical Campus, Cambridge CB2 0QH, UK (C.H.H). Correspondence and requests for materials should be addressed to R.J.R. (email: rjr27@cam.ac.uk).

Hunter syndrome, also known as Mucopolysaccharidosis type II (MPS II, OMIM 309900)[1], is an X-linked lysosomal storage disease caused by genetic deficiency of the enzyme iduronate-2-sulfatase (IDS, EC 3.1.6.13), required for the step-wise degradation and recycling of complex glycosaminoglycans (GAGs). IDS belongs to the sulfatase family of enzymes and catalyses hydrolysis of the C2-sulfate ester bond at the non-reducing end of 2-O-sulfo-α-L-iduronic acid residues in dermatan sulfate and heparan sulfate. Loss of IDS activity leads to abnormal accumulation of GAGs in multiple tissues and organs, resulting in progressive cellular and multi-organ dysfunction.

Signs and symptoms of Hunter syndrome include coarse facial features, stiff joints, skeletal abnormalities, hepatosplenomegaly, cardiovascular and respiratory disorders, developmental delay and deteriorating intellectual function. In general, the disease progression follows one of two forms. 'Severe' early-onset disease becomes apparent within 2–4 years of age, patients exhibit behavioural disturbances and progressive intellectual disability concomitant with neurodegeneration; death occurs before adulthood. In contrast, patients with the 'mild' late-onset form preserve normal or slightly impaired intelligence and survive into late adulthood[2]. The incidence of Hunter syndrome is ~1 in 100,000 male live births, but varies considerably between different populations[3,4]. Sporadic cases have been reported in females.

Over 500 mutations at the *IDS* locus (Xq28) have been identified, including complex rearrangements, insertions/deletions, splicing defects and missense/nonsense point mutations. This molecular heterogeneity results in a broad spectrum of disease phenotypes, and combined with the rarity of individual mutations, the fact that most mutations are private/familial, and the lack of standardized severity reporting, genotype/phenotype correlations have been difficult to establish[5,6]. A notable exception to this is that patients harbouring gross structural changes in the *IDS* gene, caused by deletions or gene–pseudogene recombination events, usually develop severe forms of Hunter syndrome[7]. Prediction of the clinical phenotype can also be hindered by effects of other mutations in multiple genes that may contribute to the spectrum of disease[8]. However, differences in measured level of GAG synthesis (presumably also affected by genotype) can be harnessed to improve the prediction of disease phenotype[9].

There is no curative treatment for Hunter syndrome that is currently approved, and management is usually symptomatic and supportive. Enzyme replacement therapy (ERT)[10] with recombinant human IDS (Elaprase, Shire Pharmaceuticals)[11] is effective in ameliorating peripheral symptoms associated with the 'mild' disease form, with recipients showing improved excretion of GAGs in urine, reduced liver and spleen size and some bone remodelling. However, ERT cannot prevent or reverse the characteristic cardiac and neurological deterioration of the 'severe' form, and for many patients treatment remains palliative[12,13]. Pharmacological chaperone therapy may be a useful alternative strategy for folding mutants with residual catalytic activity, as has been demonstrated for other lysosomal storage disorders[14].

Here we report the X-ray crystal structure of human IDS with a sulfate ion bound covalently in the active site. Despite low sequence identity, the overall fold and domain organization bears a striking core resemblance to other human sulfatases. This structure generates insight into the catalytic mechanism and provides a framework for better understanding the consequences of disease-associated missense mutations.

## Results

**Protein characterization**. Clinical-grade IDS was provided by Shire Pharmaceuticals in the form of a ~76 kDa glycoprotein, recombinantly expressed in HT-1080 human fibroblasts, and purified following secretion into growth medium. Upon denaturation and treatment with PNGase F, IDS can be deglycosylated to a single ~58 kDa species (Supplementary Fig. 1a). To verify that glycosylated samples for crystallization were catalytically active, assays were performed with the fluorogenic substrate 4-methylumbelliferyl-α-L-iduronide-2-sulfate (MU-αIdoA-2S, Supplementary Fig. 1b). Using this system, we report a specific activity of 36 μmol h$^{-1}$ mg$^{-1}$ at room temperature (293 K). This is comparable to the commercially reported activity of Elaprase (46–74 μmol h$^{-1}$ mg$^{-1}$), as determined with heparin disaccharide substrate at 310 K.

**Overall structure and post-translational modifications**. The crystal structure of IDS was refined at 2.3 Å resolution. Extensive crystal optimization was required to obtain the best possible diffraction data, because loose crystal packing in the *ab* plane of the crystal (Supplementary Fig. 2) led to varying levels of anisotropy (Table 1). The enzyme is monomeric with a single-polypeptide chain in the asymmetric unit. IDS adopts a compact, globular α/β sandwich fold, comprising residues 34–550 (Fig. 1). The first 33 residues, encoding the N-terminal signal peptide and propeptide, are cleaved during enzyme secretion and therefore are not present in the structure[15]. Lack of interpretable electron density prevented the modelling of one loop region (residues 444–453). A second shorter loop (residues 386–392) is very poorly ordered, but modelled with high local B-factors into weak electron density.

Traditionally, the IDS main chain has been described as two subdomains, SD1 and SD2, corresponding to fragments resulting from lysosomal proteolytic processing events[16]. The N-terminal SD1 (residues 34–443) comprises the 'heavy' 42 kDa chain, and the C-terminal SD2 (residues 455–550) corresponds to the 'light' 14 kDa chain (Fig. 1a). The lack of visible electron density in the unmodelled loop between the two subdomains (residues 444–453) is consistent with a disordered region that is susceptible to proteolytic cleavage. In our structure this loop, although disordered, is likely to remain intact as the protein is purified from conditioned medium and has not trafficked to lysosomes. This is consistent with the presence of a single-band visible by SDS–PAGE following denaturation and deglycosylation (Supplementary Fig. 1a).

SD1, which contains the catalytic core, is characterized by a central 7-stranded β-sheet (one antiparallel and six parallel strands) surrounded by ten α-helices/helical turns and five short β-strands. The C-terminal SD2 consists of a twisted, 4-stranded antiparallel β-sheet capped by a short C-terminal α-helix and flanked by five helical turns. Together, SD1 and SD2 form a large hydrophobic packing interface with a buried area of ~1,850 Å$^2$. Hence, they remain stably associated and together form the major 58 kDa 'intermediate' form seen here which is enzymatically active (Supplementary Fig. 1b). At the C-terminus, an 18-residue loop terminates in a short α-helix, which packs against the hydrophobic region of the SD2 β-sheet.

Out of the six cysteines in the mature IDS polypeptide, four are observed to form disulfide bonds. The C171–C184 bridge links short β-strands near the putative substrate recognition site, while the C422–C432 linkage participates in stabilization of an extended (> 30 residue) solvent-exposed loop region. Both disulfide bonds are functionally important, as mutation of any of these cysteines causes Hunter syndrome (Supplementary Table 1). One free cysteine is found at the tip of an 18-residue loop connecting strand β1 of the major β-sheet and N-terminal α-helix 1. The other, C84, is post-translationally modified as a key catalytic residue in the active site.

**Table 1 | Data collection and refinement statistics.**

| | IDS-native (5FQL) | IDS-GdCl$_3$ | IDS-NaI |
|---|---|---|---|
| *Data collection* | | | |
| Wavelength (Å) | 0.9763 | 1.475 | 0.9763 |
| Space group | $P\,3_1\,1\,2$ | $P\,3_1\,1\,2$ | $P\,3_1\,1\,2$ |
| Cell dimensions | | | |
| $a, b, c$ (Å) | 71.1, 71.1, 285.9 | 68.8, 68.8, 289.0 | 72.6, 72.6, 286.4 |
| $\alpha, \beta, \gamma$ (°) | 90, 90, 120 | 90, 90, 120 | 90, 90, 120 |
| Resolution (Å) | 35.52–2.30 (2.39–2.30)* | 80.0–3.28 (3.36–3.28) | 62.90–3.50 (3.70–3.50) |
| $R_{merge}$ | 0.080 (0.944) | 0.166 (2.67) | 0.075 (1.17) |
| $<I/\sigma I>$ | 12.4 (1.9) | 17.4 (1.3) | 18.2 (2.6) |
| CC(1/2) | 0.997 (0.781) | — | — |
| Completeness (%) | 96.2 (97.7) | 99.0 (92.1) | 99.2 (99.8) |
| Redundancy | 5.3 (5.2) | 15.8 (10.0) | 11.5 (11.7) |
| Anisotropic ΔB (Å$^2$) | 38 | 117 | 134 |
| | | | |
| *Refinement* | | | |
| Resolution (Å) | 35.53–2.30 (2.38–2.30) | — | — |
| No. reflections | 35,738 (3,605) | — | — |
| $R_{work}/R_{free}$ | 0.181/0.210 | — | — |
| No. atoms (excl. H) | | | |
| Protein | 4,269 | — | — |
| Water | 213 | — | — |
| B-factors | | | |
| Protein | 67.7 | — | — |
| Water | 57.3 | — | — |
| R.m.s.d. | | | |
| Bond lengths (Å) | 0.004 | — | — |
| Bond angles (°) | 0.655 | — | — |

r.m.s.d., root mean squared deviation.
*Values in parentheses are for highest-resolution shell. One crystal was used per structure.

The IDS sequence contains eight putative N-linked glycosylation sites (Fig. 1a). Continuous electron densities were observed adjacent to all asparagine residues involved in N-linked glycosylation, and at least one N-acetylglucosamine moiety linked to the nitrogen of the asparagine side chain was built at each predicted site.

**Comparison to other sulfatase enzymes.** To date, 17 genes encoding human sulfatases have been identified. Amino acid sequences vary significantly amongst members of the sulfatase family, and pairwise alignment of IDS with other human sulfatases reveals only ∼20% sequence identity, with greater conservation in N-terminal regions (Supplementary Fig. 3). X-ray crystal structures have been determined of human sulfatases arylsulfatase A (ARSA)[17], arylsulfatase B (ARSB)[18] the membrane-associated estrone or steroid sulfatase (STS, also known as arylsulfatase C)[19], galactosamine-6 sulfatase (GALNS)[20] and N-sulfoglucosamine sulfohydrolase (SGSH)[21]. Deficiencies of these enzymes also lead to rare inherited disorders: metachromatic leukodystrophy, Maroteaux–Lamy syndrome (MPS VI), X-linked ichthyosis, Morquio A syndrome (MPS IV A) and Sanfilippo A syndrome (MPS III A), respectively.

Structural alignments reveal significant topological similarity between N-terminal regions of IDS and other sulfatases, concentrated around the central β-sheet in SD1 (Fig. 2a, 174 residues with backbone Cα RMSD<2.0). Beyond this, poor conservation of surface loops and divergent C-terminal sequences prevent main-chain superposition, but the 4-stranded β-sheet in SD2 is a feature shared by all human sulfatases. Nevertheless, several large differences are observed. The central β-sheet at the core of IDS is formed by seven strands in contrast to ten for O-sulfatases ARSA, ARSB, STS and GALNS, more closely resembling the 8-stranded core of SGSH, an N-sulfatase. This

has consequences for the shape of the putative substrate-binding region. In IDS, the cleft is relatively wide and shallow, while in other sulfatases it forms a narrower cavity, deepening towards the catalytic site (Fig. 2b–e). In arylsulfatases A and B this cavity is extended by three additional strands of the central β-sheet; in STS, the two transmembrane helices emerge from the globular domain directly adjacent to the substrate-binding cleft, forming a deep-binding pocket (Fig. 2e). Differences are also observed in the C-terminal regions; most notably the very extensive C-terminal loops or β-meander motifs present in other sulfatases are absent from IDS. In ARSA, ARSB and GALNS these C-terminal extensions help to define the substrate-binding pockets. In IDS, the four antiparallel strands comprising the SD2 β-sheet are considerably longer than those of other sulfatases, and hence a greater contribution to the shape of the substrate-binding cleft comes directly from SD2.

**Substrate-binding cleft and active site.** Despite highly divergent sequences and the diversity of substrates processed by these enzymes, the active-site residues (red stars in Supplementary Fig. 3) are highly conserved and are almost completely superimposable in all six sulfatase structures (Fig. 2f), providing strong support for a common catalytic mechanism. The active site of IDS is located in a cleft comprising many basic residues, conferring a positive electrostatic surface potential consistent with binding negatively charged polymeric substrates (Figs 3a and 1b). The catalytic residue C84 is situated at the N-terminus of helix 2, opposite the central β-sheet. In all eukaryotic sulfatases, the thiol group of this residue is oxidized to Cα-formylglycine (FGly, 2-amino-3-oxopropionic acid), an essential modification required for catalytic activity[22]. This modification is performed by FGly-generating enzyme (FGE), which recognizes the highly conserved sequence motif **C**xPSR during translocation of the

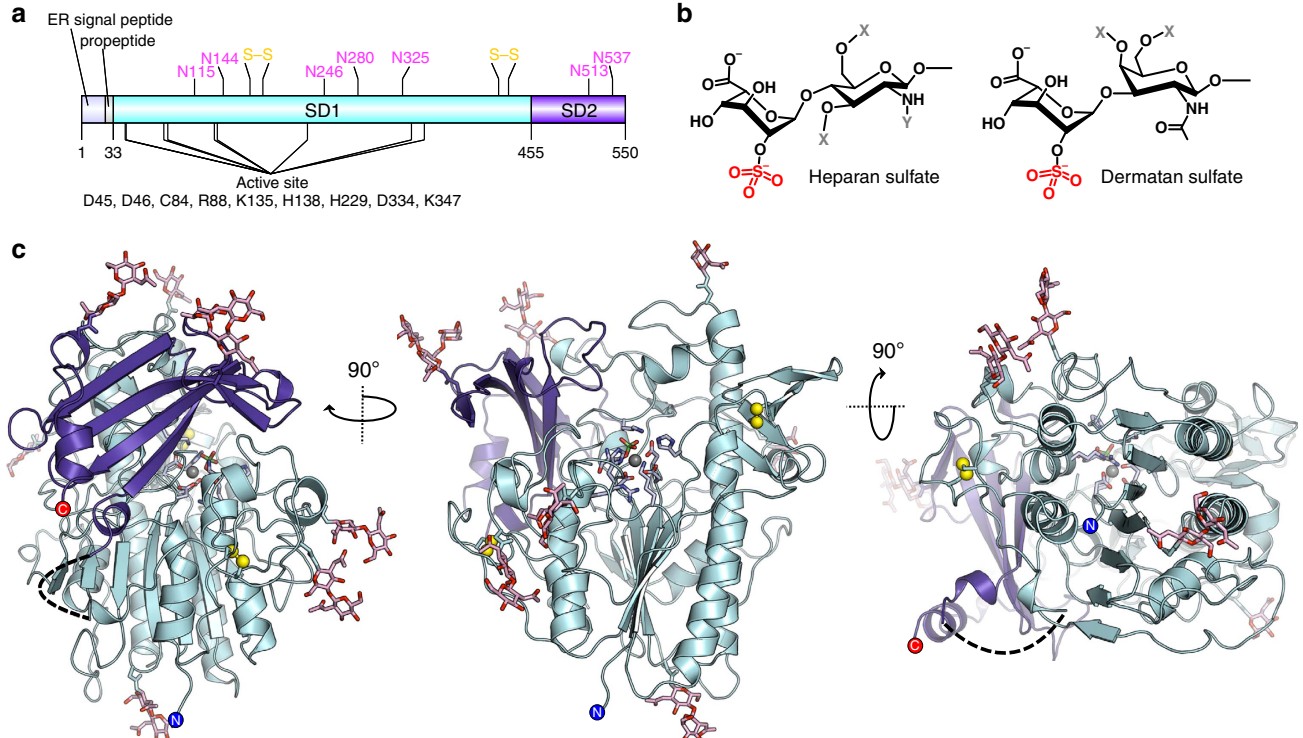

**Figure 1 | X-ray crystal structure of IDS.** (**a**) Annotated diagram of the IDS sequence, showing the ER signal peptide and propeptide (grey), subdomains (SD) 1 and 2 (light blue and purple, respectively), active-site residues (black), disulfide bridges (yellow) and N-linked glycosylation sites (pink). (**b**) Natural IDS substrates heparan sulfate (with α1-4 linkage) and dermatan sulfate (α1-3 linkage). The sulfate group removed by IDS is indicated (red). The residue at the +1 position can be either D-glucosamine or D-galactosamine, with variable substituents at C2$_N$, C3, C4 and C6 positions (X = H/SO$_3^-$; Y = H/SO$_3^-$ / COCH$_3$) (**c**) Ribbon diagram of IDS coloured by subdomain, shown in three orthogonal views. Disulfide bonds (yellow spheres), calcium ion (grey sphere) and glycans (pink sticks) are shown. The active site is indicated by grey sticks.

nascent (and unfolded) polypeptide chain into the endoplasmic reticulum (ER) lumen[23,24]. FGly functions catalytically as an aldehyde hydrate (FGly aldehyde-hydrate, FGH) with two geminal hydroxyl groups; here we observe what we have interpreted as a covalent sulfate-ester intermediate (FGly sulfate, FGS) in the IDS active site (Fig. 3b), which could originate from natural substrate encountered in cells during expression. An alternative interpretation that this is a phosphate picked up from the storage buffer, as proposed for SGSH[21], cannot be excluded on the crystallographic evidence. However, the higher stability of sulfate esters makes this the more likely interpretation.

Extra electron density for a metal ion was observed next to the modified FGS residue. This was modelled as Ca$^{2+}$, coordinated by one nitrogen and five oxygen atoms with approximate octahedral geometry: one oxygen each from the side chains of D45, D46 and D334; two sulfate oxygens from FGS84 and one nitrogen from the side chain of H335. (Fig. 3c). The metal–oxygen bond distances range from 2.3 Å to 2.5 Å, corresponding to the expected values for calcium[25]. The second carboxylate oxygen of D334 forms an additional longer interaction of 3.3 Å. Successive temperature factor refinements for bivalent cations Ca$^{2+}$, Mn$^{2+}$ and Mg$^{2+}$ resulted in values of 55.2 Å$^2$, 65.4 Å$^2$ and 39.7 Å$^2$, respectively; the B-factor for a Ca$^{2+}$ ion correlates best with the average B-factor (52 Å$^2$) of the coordinating side chain atoms[25]. Calcium was also identified in the metal-binding pocket of human ARSB[18], STS[19], GALNS[20], SGSH[21] and *Pseudomonas aeruginosa* arylsulfatase (PAS)[26] whilst in human ARSA, both magnesium and calcium have been described[17,27]. With its lower Lewis acidity, Ca$^{2+}$ is more consistent than Mg$^{2+}$ with the stability of the observed

sulfate-ester modification of formylglycine: in related structures, the identification of either a sulfate ester (ARSB[18], STS[19]) or phosphate ester (ARSA[17,27], SGSH[21]) modification correlates with the identification of the metal ion as Ca$^{2+}$. The presence of divalent metal ions in sulfatases is thought to be essential for active-site stabilization and sulfate-ester formation[28].

Nine out of ten IDS active-site residues are highly conserved among sulfatases: D45, D46, C84, R88, K135, H138, H229, D334 and K347. The histidine residue at position 335 is replaced in most sulfatases by asparagine, and by glutamine in STS (Supplementary Fig. 3). Side chains of conserved H138 and H229 likely play important roles in catalysis, making direct contacts with the free geminal hydroxyl of FGS and a sulfate oxygen (OS3) of FGS, respectively. The positively charged groups of R88, K135 and K347 make several structural interactions within the catalytic core. R88 stabilizes the orientation of metal-binding residues D45 and D334 and interacts with the free geminal hydroxyl of FGS; K135 also interacts with D45 to correctly orient the side chain for metal-binding. D46 is positioned for metal-binding via interactions with K347 and W337. The sulfate moiety of FGS is stabilized by hydrogen bonds with the side chains of D46, K135, D334 and K347 and, via a water molecule, to the side chains of Y348 and K479 (Fig. 3d).

Attempts to obtain an enzyme–substrate complex structure by co-crystallization or soaking with a variety of ligands were not successful. Nevertheless, to gain insight into possible modes of substrate binding, we performed docking experiments with 2-sulfo-iduronic acid, heparan sulfate disaccharides and a dermatan sulfate disaccharide, restraining the position of the scissile sulfate group adjacent to the catalytic FGH residue with appropriate geometry for attack (Fig. 3e). This demonstrates

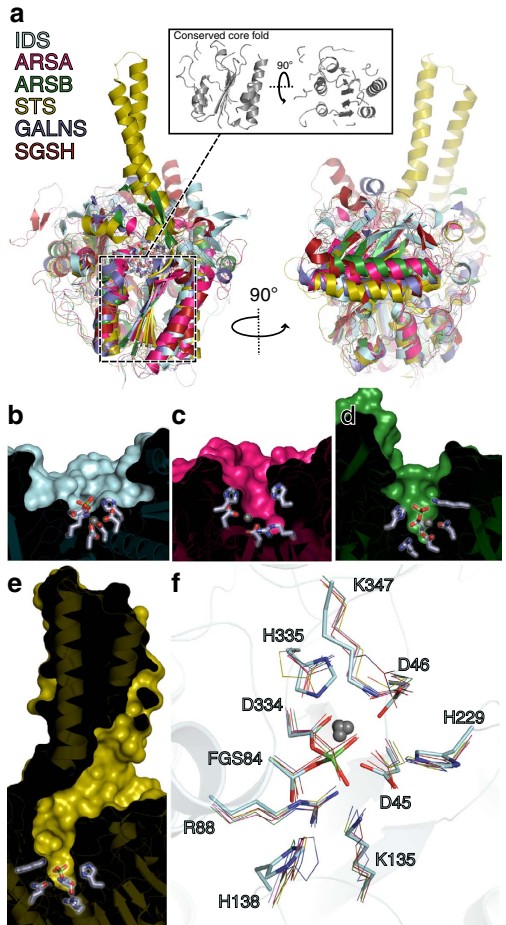

IDS
ARSA
ARSB
STS
GALNS
SGSH

Conserved core fold

90°

**Figure 2 | Structural alignment of IDS with other human sulfatase enzymes.** (**a**) Secondary structure-based superposition of IDS (blue), ARSA (pink), ARSB (green), STS (yellow), GALNS (slate) and SGSH (dark red) in two orthogonal views. Active-site residues are shown as grey sticks. Inset: core structural elements shared by all human sulfatases, showing only conserved regions with backbone Cα RMSD < 2.0 Å. (**b**–**e**) Cutaway diagrams of the substrate-binding cleft of IDS, ARSA, ARSB and STS, coloured as above. Active-site residues are shown as grey sticks. (**f**) Least squares superposition of conserved active-site residues, coloured as above. The metal ions are shown as grey spheres.

that it is possible to place the substrate in a favourable orientation consistent with our mechanistic understanding. The 2S-IdoA monosaccharide binds in a low energy $^4C_1$ conformation, making additional hydrogen bonds to R273, N167 and Y348 from SD1, and K479 from SD2. The substrate-binding cleft is large enough to accommodate both the heparan sulfate (α1–4 linkage) and dermatan sulfate (α1–3 linkage) disaccharides with different substituents at the +1 position: H226, K479 and K135 can make polar contacts with additional OH, $SO_3^-$ or $COCH_3$ groups. Different ring puckers of 2S-IdoA (ref. 29) may also be tolerated by the binding site. An analysis of mean B-factors per residue reveals that the residues proximal to the scissile sulfate are well ordered, indicating that the active site is likely preorganised. However, K479 and D269 are likely flexible and may change conformation to accommodate the diverse and heterogeneous substrates processed by this enzyme.

## Discussion

Two mechanisms for sulfate-ester hydrolysis have been proposed based on crystal structures of human ARSA, human ARSB and

*Pseudomonas* arylsulfatase (PAS), and studies on the enzyme kinetics of active-site mutants[30]. Most recent experimental evidence supports the transesterification-elimination (TE) mechanism, in which the catalytically relevant species is an FGly aldehyde-hydrate (FGH) with a second geminal hydroxyl. The arrangement of the IDS active site is compatible with the consensus TE mechanism, hence we propose a similar mechanism for IDS in which FGH84 performs an $S_N2$ nucleophilic attack on the sulfur atom to generate the covalent FGS adduct, followed by elimination of bound sulfate and rehydration of the resultant FGly aldehyde. In particular, H229, H138, D334 and K347 are likely to play active roles in proton transfer or activation of water during the reaction cycle, as proposed for other sulfatases.

Interestingly, we observe a sulfate ion bound in the active site, similar to the observed 'resting state' of ARSB and STS. The covalent linkage is unambiguous, with a distance of 1.5 Å between the sulfur atom and the FGS ester oxygen. Attempts to refine a non-covalently bound sulfate ion adjacent to an FGH residue, as seen in the high-resolution PAS structure, did not agree with the electron density present in the IDS structure (Supplementary Fig. 4). Due to the dependency on general acid/base catalysis, it is likely that sulfate elimination will only occur efficiently at the optimum lysosomal pH of ∼ 4.8, and not in the storage buffer (pH 7.5) or crystallization buffer (pH 6.1). Indeed, despite this observed sulfate ion, IDS samples were enzymatically active at pH 5.0 (Supplementary Fig. 1b).

To date, 542 pathogenic mutations in the *IDS* gene have been identified in patients with Hunter syndrome (Supplementary Table 1). The mutation spectrum is diverse and ranges from small point mutations, deletions and insertions to truncations, large *IDS* gene deletions and complex rearrangements. A large proportion of mutations associated with Hunter syndrome are missense mutations randomly distributed throughout the protein sequence, with 217 amino acid substitutions described at 128 positions (Fig. 4). These are likely to result in protein misfolding, catalytic inactivation, premature degradation or failure of lysosomal targeting. However, at least five of these are likely to be false positives, as they are found in the Exome Aggregation Consortium database[31] at a frequency too high to be consistent with a role in this rare disease (Supplementary Table 1).

The structure of IDS provides detailed insight into the consequences of many disease-associated mutations. Several mutations directly affect the catalytic core of the enzyme, either by direct substitution of key active-site residues, interference with metal ion coordination, indirect active-site destabilization/distortion or corruption of the CxPSR motif necessary for the essential FGly modification (Fig. 4). Such changes are likely to eliminate or severely reduce catalytic activity with grave phenotypic consequences; for example R88H/C/P/L/G/S mutations and K347I/Q/T/E mutations often lead to severe, early-onset disease.

Outside the active site, a small number of mutations are located on the solvent-accessible surface of the enzyme. These include substitutions that alter local surface charge (E125V) and others that interfere with N-linked glycosylation by either removal of existing sites (N115I/Y, S117Y), or aberrant introduction of new glycans (T130N) (Fig. 4). As well as causing misfolding, incorrect glycosylation may interfere with lysosomal targeting by impaired binding to the mannose-6-phosphate receptor.

In contrast to those affecting the enzyme surface, nearly 90% of pathogenic missense mutations involve amino acids that are buried within the fold of IDS (Supplementary Table 1). The expected severity of these mutations is variable and consequences range from local destabilization and misfolding to global unfolding, leading to premature degradation. In the tightly

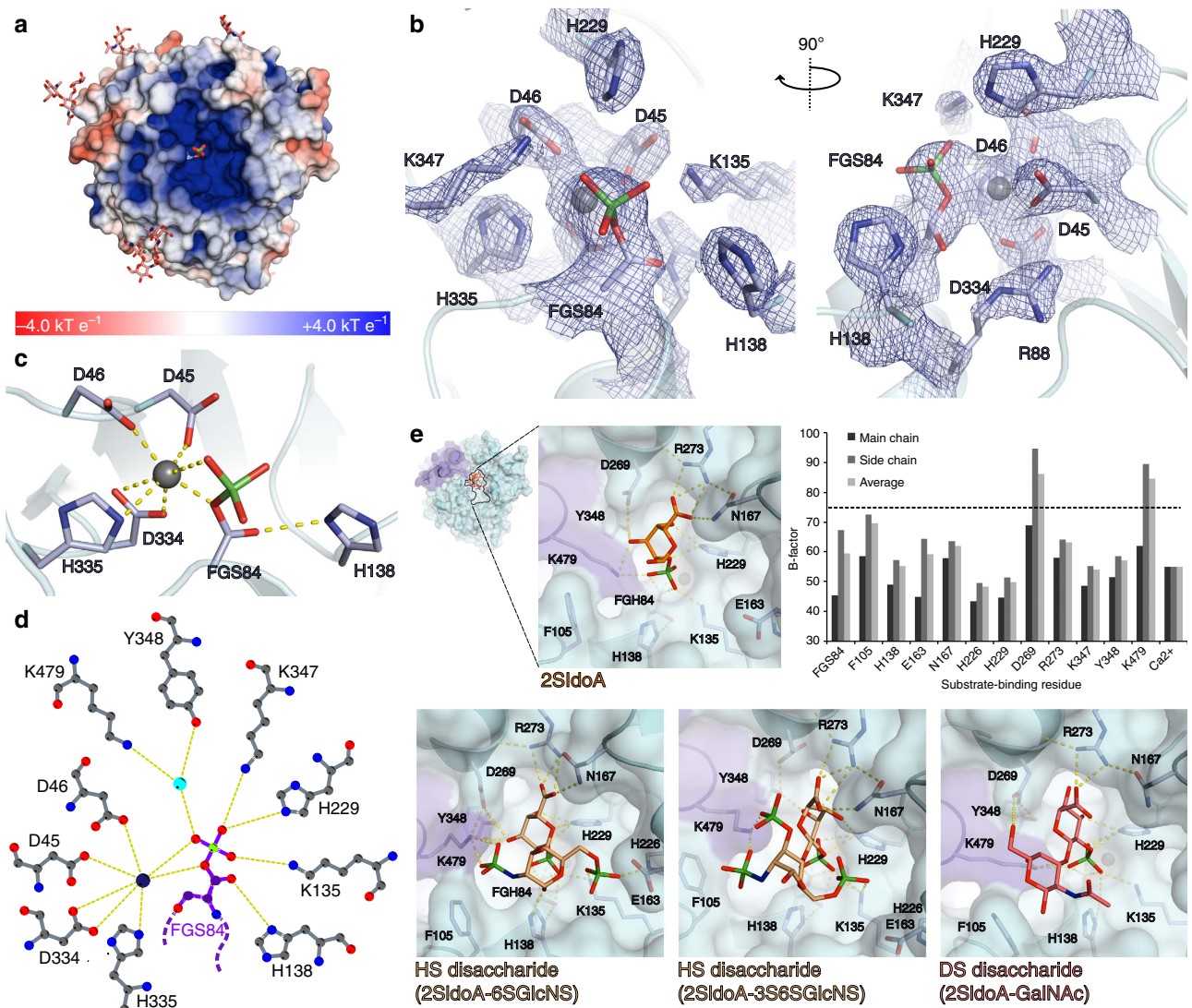

**Figure 3 | Details of IDS active site and substrate-binding cleft.** (**a**) The substrate-binding cleft is coloured by electrostatic potential at the solvent-accessible surface from red (negative, $-4.0\,\mathrm{kTe}^{-1}$) to blue (positive, $+4.0\,\mathrm{kTe}^{-1}$). Electrostatic potential was calculated using a pH value of 4.8 when assigning side chain protonation. The catalytic, modified FGS84 residue is shown as sticks. (**b**) Two orthogonal views of electron density (blue mesh, $2mF_O-DF_C$, contoured at $0.35\,\mathrm{e}^-\,\mathrm{\AA}^{-3}$) for IDS active-site residues (grey sticks) and the $Ca^{2+}$ ion (grey sphere). (**c**) Details of the $Ca^{2+}$ ion coordination (yellow dashed lines) in the IDS active site. (**d**) Schematic representation of amino acid side chain interactions with the covalently-modified FGS84 residue. Hydrogen bonds (yellow dashed lines) are shown. (**e**) Docked structures of 2-sulfo-iduronic acid, two variably sulfated heparan sulfate disaccharides and a dermatan sulfate disaccharide, showing details of the putative substrate-binding region. Hydrogen bonds (dashed yellow lines) are indicated. For solvent-accessible residues situated in the substrate-binding cleft, a graph of average B-factor per residue is shown. The mean B-factor for all IDS non-hydrogen atoms is indicated with a dotted line.

packed hydrophobic core surrounding the central β-sheet, several such mutations (L41P, L67P, L72P, L73F, L221P, L314P/H, L403R and L410P) would introduce main-chain geometry distortion or buried steric clashes, disrupting the local secondary structure in the core of SD1 (Fig. 4). In SD2, the guanidinium group of R468 participates in a buried hydrogen bonding network and hydrophobic stacking interactions stabilizing several nearby loops, helical turns and β-strands (Fig. 4, Supplementary Fig. 5). Mutation of this residue (R468Q/L/W/G/P) is associated with a range of Hunter disease phenotypes from mild to severe.

In general, proteins with folding defects will be retained by the ER quality control machinery, preventing trafficking through the Golgi apparatus and final delivery to lysosomes[32]. In experiments with COS cells, transiently expressed P86L/R and P480L/Q

mutants did not efficiently traffic to lysosomes[33]. However, misfolding does not always lead to catalytic inactivation, and several IDS missense mutants with residual enzyme activity have been identified[33–37]. Such mutants could be promising candidates for treatment by pharmacological chaperone therapy (PCT). PCT relies on use of small molecules which selectively bind and stabilize target proteins, shifting the folding equilibrium towards functional material, thus alleviating ER retention and increasing lysosomal delivery. This therapeutic approach is under active development for several other lysosomal storage disorders including Pompe[38], Fabry[39], Gaucher[40,41] and Krabbe disease[42,43]. As well as the IDS active site, several other surface pockets with different electrostatic potentials are present which may be exploited for future chaperone development (Supplementary Fig. 6).

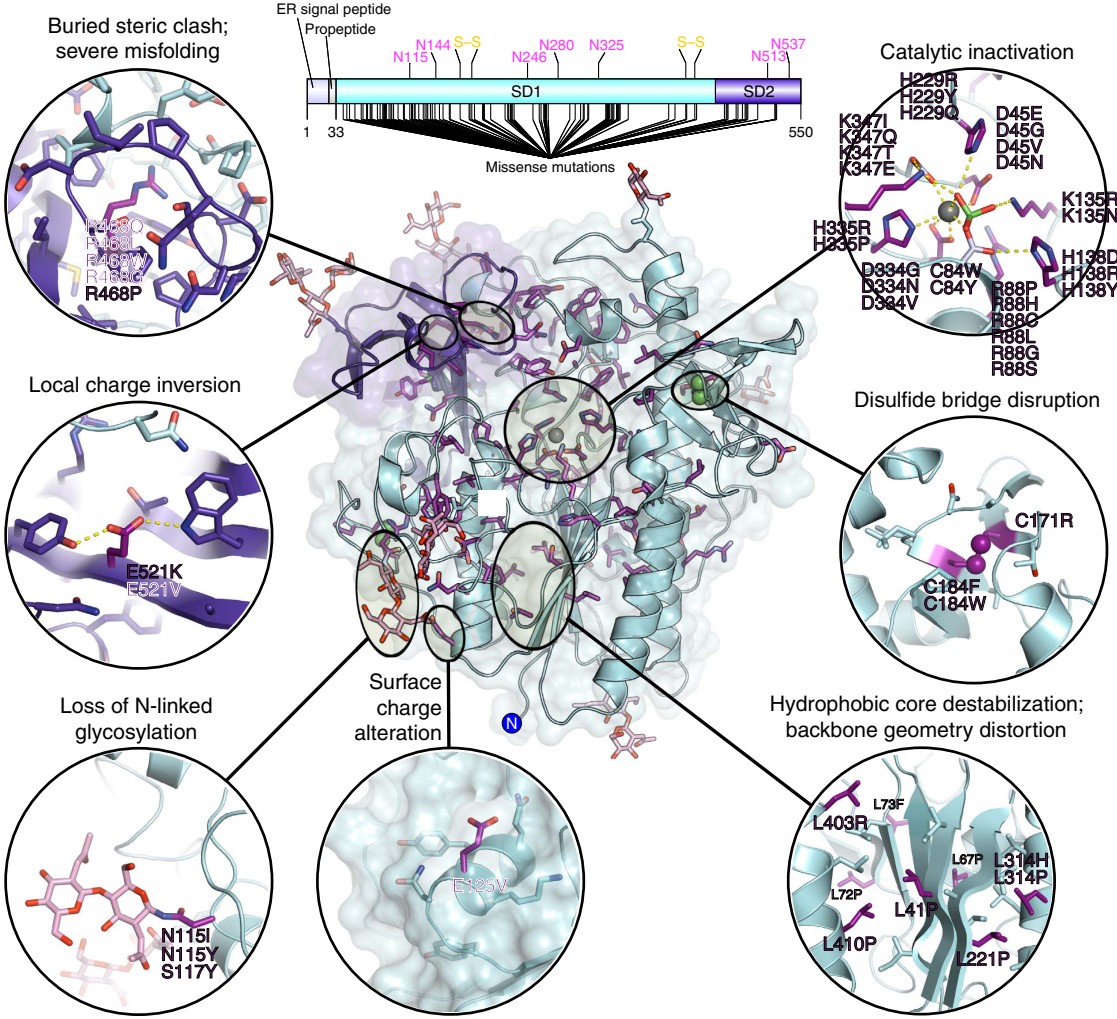

**Figure 4 | Disease-associated mutations of IDS.** (Top) Annotated diagram of the IDS protein, indicating positions of all known Hunter syndrome missense mutations. All residues that are mutated are shown as sticks (purple) on the structure of IDS, coloured as above. Different mutations have different consequences; for a selection of these a more detailed view is provided (Insets) showing the surrounding region of the structure that is perturbed.

IDS variants with disruptive mutations such as complete deletions or large rearrangements will not be amenable to PCT, and patients carrying such mutations are also more susceptible to the development of neutralizing antibodies that reduce the efficacy of ERT[44]. Knowledge of the structure of IDS could also facilitate structure-based de-immunization[45,46] of the protein used in ERT, to remove T-cell epitopes while preserving a stable structure.

The structure of IDS described here provides a much-needed snapshot of the enzyme defective in Hunter syndrome, delivering a detailed atomic framework for the rationalization and further understanding of disease-causing mutations, and supplying 'blueprints' for the future development of new PCT agents or less immunogenic protein for ERT. In addition, details of the active site support a likely catalytic mechanism shared by other sulfatases, adding to our understanding of this family of enzymes.

## Methods

**Deglycosylation experiments.** IDS protein in 100 mM NaCl, 10 mM Tris · HCl pH 7.5 was denatured by the addition of urea up to 2.5 M final concentration. Samples were treated with PNGase F (Sigma) as directed by the manufacturer (5 h, 310 K) before analysis by reducing SDS–PAGE.

**Enzyme activity assay.** Fluorogenic assays were performed based on the methods of Voznyi et al.[47] and Tolun et al.[48]. In brief, the sulfate group on substrate 4-methylumbelliferyl-α-L-iduronide-2-sulfate (MU-αIdoA-2S) is hydrolysed by IDS, upon which the desulfated product 4-methylumbelliferyl-α-L-iduronide (MU-αIdoA) becomes a substrate for reporter enzyme α-iduronidase, which cleaves MU-αIdoA to generate fluorescent product 4-methylumbelliferone.

Substrates and reporter enzymes were obtained from Moscerdam Substrates. Reactions of 50 µl comprised 0.3 mM MU-αIdoA-2S, 1.3 nM IDS (100 ng ml$^{-1}$) and excess α-iduronidase reporter enzyme in 0.1 M sodium acetate pH 5.0, 10 mM lead acetate, 0.01% Tween-20. Reactions were incubated at room temperature (293 K) for 60 min before termination with 200 µl stopping buffer (0.5 M Na$_2$CO$_3$ pH 10.7, 0.025% Triton X-100). Liberated 4-methylumbelliferone was quantified by measuring fluorescence ($\lambda_{ex}$ = 365 nm; $\lambda_{em}$ = 445 nm) with a SpectraMax M2 plate reader (Molecular Devices).

**Protein preparation and crystallization.** Purified recombinant human IDS (Elaprase) was kindly provided by Shire Pharmaceuticals. Crystallization trials were carried out with both glycosylated and deglycosylated protein, but considering that no improvement in diffraction quality was observed after deglycosylation and that the two crystal forms were isomorphous, the glycosylated material was used for structure determination. IDS was buffer-exchanged by dialysis (10-kDa molecular mass cut-off; Thermo Scientific) into 100 mM NaCl, 10 mM Tris HCl pH 7.5 and concentrated to 10.0 mg ml$^{-1}$ using Amicon Ultra 10-kDa molecular mass cut-off centrifugal concentrators (Millipore). Crystals were grown at 20 °C by hanging drop vapour diffusion. In total, 1.0 µl of protein was mixed with 1.0 µl reservoir solution (0.1 M MES-NaOH pH 6.1, 1.0 M LiCl and 20% wt/vol PEG 6 K) and equilibrated against 1.0 ml reservoir solution. Crystals were cryoprotected by transferring to a drop of cryo solution (equilibrated reservoir solution increased to 30% wt/vol PEG 6 K and supplemented with 20% wt/vol ethylene glycol) for 30 s before flash-cooling in liquid nitrogen.

**Crystal soaks for experimental phasing.** For the NaI-derivative data set, crystals were soaked sequentially in cryo solution (above) supplemented with 0.4 M NaI (1 min) and 1.0 M NaI (5 min). For the GdCl$_3$-derivative data set, crystals were prepared by soaking in cryo solution supplemented with 50 mM GdCl$_3$ (30 min). Crystals were then flash-cooled as above.

**Crystallographic structure determination and analysis.** Data collection and processing statistics are detailed in Table 1. All data sets were collected at a temperature of 100 K. Diffraction data were recorded at Diamond Light Source beamline I03 on a Pilatus 6M detector (Dectris). The native data set was collected at $\lambda = 0.9763$ Å over a 360° oscillation range with $t = 0.15$ s and $\Delta\Phi = 0.2$ ° per image. Diffraction data were processed using the XIA2 pipeline[49] implementing MOSFLM[50] for indexing and integration, POINTLESS[51] for point group determination and AIMLESS[52] for scaling and merging. iMOSFLM[50] was also used for manual indexing and integration. Resolution cut-off was decided by $CC_{1/2} > 0.5$ and $\langle I/\sigma I \rangle \geq 1.8$ in the outer resolution shell[53]. The NaI-derivative data set was collected as above; an initial attempt to collect data at the iodide peak wavelength suffered from excessive radiation damage. The GdCl$_3$-derivative data set was collected at $\lambda = 1.475$ Å.

SAD phases useful to about 3.8 Å resolution were determined for the GdCl$_3$-derivative crystal with phenix.autosol[54], which found 4 Gd sites. The resulting electron density was not interpretable but allowed docking of known sulfatase structures to aid further interpretation; the best agreement was obtained with PDB entry 3B5Q, a putative sulfatase from *Bacterioides thetaiotaomicron* VPI-5482. Electron density cut out from the SAD map, using the docked sulfatase structure as a mask, was used to solve the native and iodide-soak data by molecular replacement in Phaser[55]. A substructure of 11 iodide sites was then determined using SAD log-likelihood-gradient maps[56], starting from the electron density model. At this point, initial phase information, a molecular mask, and operators relating the positions of IDS in the three crystals were available, allowing averaging to be carried out with phenix.multi_crystal_average[57]. An iterative process of cycles of automated model-building with phenix.autobuild[58], followed by determination of updated substructures (ultimately 8 sites for the GdCl$_3$ derivative and 13 sites for the iodide soak) and multi-crystal averaging, eventually led to a model comprising 405 of 517 residues, with $R_{free} = 0.42$.

All further model building and refinement was performed iteratively using COOT[59] and phenix.refine[60] (Table 1). The Protein Data Bank file and CIF restraints for the non-standard FGS84 residue were generated using eLBOW[61]. Hydrogens were refined in the riding position. MolProbity[62] was consulted throughout the refinement process, at the end of which 95.4% of residues were in the favoured Ramachandran region and none were outliers. Estimation of buried surface area in the IDS structure was carried out using PISA[63]. For the electrostatic potential calculations, partial charges were assigned using the PDB2PQR server[64], implementing PROPKA[65] to estimate side chain pKa values. Electrostatic surfaces were calculated using APBS[66] and structural figures were rendered using PyMOL (Schrödinger LLC). Two-dimensional hydrogen bonding diagrams were created using LigPlot+ (ref. 67).

**Docking.** To prepare the protein model for docking, the sulfated FGS84 residue was replaced with FGH (Protein Data Bank code DDZ: 3,3-dihydroxy L-alanine) and waters and glycans were removed. Ligand PDB files for 2-sulfo-iduronic acid and various heparan sulfate and dermatan sulfate disaccharides were generated with the GLYCAM-Web GAG builder (http://glycam.org) and minimized with PRODRG[68]. All docking was performed with HADDOCK 2.2 (ref. 69) using the WeNMR grid service[70]. Residues FGH84, Ca$^{2+}$, K135, K347 and H229 were specified as active, with passive residues defined automatically within a 6.5 Å radius of active residues. Unambiguous distance restraints were enforced to maintain the correct orientation of the nucleophile (via interaction with H138) and preserve Ca$^{2+}$ coordination geometry during docking. For ligands with multiple sulfate groups, an additional restraint was provided to juxtapose the relevant iduronate-2-sulfate group close to the active site.

**Amino acid sequence alignment.** Primary sequences for known human sulfatases were retrieved from the UniProt database[71] and annotated ER signal peptides were removed. Sequences were aligned by using ClustalW[72], and graphical representations of the alignment were prepared using ALINE[73].

**Data availability.** The atomic coordinates and structure factors for IDS have been deposited in the wwPDB under accession code 5FQL. All other data are available from the corresponding author upon reasonable request.

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

## Acknowledgements

We acknowledge Diamond Light Source for time on beamline I03 under proposal MX6641. We thank Shire Pharmaceuticals for providing Elaprase (idursulfase) and Tom Terwilliger for helpful advice on multi-crystal averaging. We also thank Alexandre Bonvin for modifying HADDOCK to support the non-standard amino acid FGH (PDB code DDZ). R.J.R. is supported by a Principal Research Fellowship funded by the Wellcome Trust (Grant 082961/Z/07/Z), which also supported C.H.H. and M.D. A.Z. was supported by a Senior Research Fellowship from the British Heart Foundation (PG/09/072/27945). J.E.D. is supported by a Royal Society University Research Fellowship (UF100371). Support received from the US National Institutes of Health (grant P01GM063210 R.J.R.) is gratefully acknowledged. The research was facilitated by a Wellcome Trust Strategic Award (100140) to the Cambridge Institute for Medical Research.

## Author contributions

M.D. prepared protein samples, performed biochemical experiments, collected crystallographic data, carried out the initial refinement and co-wrote the manuscript. A.Z., P.E.S. and D.M. assisted with protein chemistry and crystallization. R.J.R. solved the structure. G.B. assisted with refinement. C.H.H. refined and analysed crystallographic data, carried out docking calculations, prepared figures and co-wrote the manuscript. P.E.S., J.E.D. and R.J.R. co-wrote and edited the manuscript.

## Additional information

**Competing interests:** The authors declare no competing financial interests.

