## [Peer review file · Nature Communications]

Reviewer #1 (Remarks to the Author):

NCOMMS-16-26546 by Demydchuk *et al.*

This manuscript describes the long-awaited crystal structure of IDS. The structure provides a platform to explain the phenotypes observed in many variants leading to Hunter syndrome, and may offer routes to the therapy of patients that have specific IDS variants with residual activity. I have these comments:

Major

- The packing shows that no crystal contacts exist along the a-axis, which is in principle physically impossible, although there are several legitimate examples in the PDB. As the glycosylated material was used for crystallization, disordered oligosaccharides could provide unspecific crystal contacts, but this is nowhere stated. Was there strong anisotropy in the data? The statistics in Table 1 appear very good for the resolution (although R_{merge} is deprecated).
- The sentence “An alternative interpretation that this is a phosphate picked up from the storage buffer...” invites mass spectrometry experiments, e.g. production of proteolytic fragments at pH where the esters are stable, followed by mass analysis.
- The biochemical reason why sulfatases have FGly is that sulfate esters are more difficult to hydrolyze than are phosphate esters. The related serine phosphatases need only a Ser in their active site because the enzyme intermediate has higher energy. Sulfatases with a Ser instead of FGly are stuck at the stable sulfated enzyme intermediate. The enzyme-sulfate intermediate in active sulfatases is resolved by the elimination reaction. This circumstance would hint towards a sulfate, as suggested by the authors, rather than a phosphate in the IDS structure. The question is why a sulfate is trapped in the first place? This should be discussed by the authors. Possibly, the Lewis acidity of Ca^{2+} is insufficient? That calcium is present in the structure is convincingly shown, but where does it come from? As far as I know, the physiological ion is Mg^{2+} , which is a stronger Lewis acid than Ca^{2+} . And since Mg^{2+} is coordinated strictly octahedrally, the distorted configuration seen in the IDS structure might be an off-pathway structure that might have enabled trapping of the sulfate ester.
- Regarding the mechanism in Fig. 3e: I have yet to find a single structure of a carboxylic acid or an alcohol bound to a metal ion in the CSD that has convincing electron density for the proton(s), but maybe the authors know of examples? It appears that if a metal ion is bound to oxygen, the proton generally dissociates, even if this means a surplus of negative charge next to the metal ion that has to be neutralized elsewhere. Hence, the top left part of the figure where FGH binds Ca^{2+} and retains its proton might be wrong. FGH might already be present as an anion, stabilized by the positive potential in the active site. While oxyanions in phosphatases and proteases have to be stabilized by many interactions (NH-groups in the oxyanion hole”, the helix dipole), the additional hydroxyl group of hydrated FGly reduces the

pKa of either hydroxyl, and metal ion binding might lead to deprotonation, so the nucleophilic anion could be a stable ground state. The positive end of the helix dipole where FGly is located will also contribute to stabilization of an anion.

- “...and by D334 donating a proton to the departing sulfate”. This is shown in the bottom right panel of Fig. 3e. Since HSO₄⁻ has a pKa of 2, it is unlikely for Asp334 to donate a proton. It is also not needed, since sulfate is a very good leaving group. I would argue that Asp334, being bound to the metal ion, is deprotonated throughout the catalytic cycle and does not need to play an active role.
- “The final rehydration of the resultant FGly aldehyde is assisted by activation of water by H229 and protonation by H138”. With “protonation” you probably mean “protonation of the hydrated aldehyde”, not the water molecule. H229 is solvent-exposed, and thus very likely always protonated at ambient pH (lysosome) unless shielded by the substrate (a model could help here, see also below). Alternatively, water may attack FGly and the proton is then removed by other waters. The Grotthus mechanism would be a feasible route for the proton to exit the active site. Since the insights into the sulfatase mechanism provided by this structure are limited, I suggest keeping its discussion at a minimum.
- Despite the presence of the sulfate, IDS was active at low pH. This could be of biological significance – inactive enzyme during transport, activation at its target location. However, the notion invites more experiments: A new structure after pre-treatment of IDS at lower pH would tell whether this hypothesis is true (the sulfate should be absent). Addition of Mg²⁺ to the crystallization experiment could replace the Ca²⁺. Alternatively, MS analysis at different pH could help (see second point above).
- The Discussion makes a strong point of the possibility that certain Hunter syndrome-causing variants of IDS can be stabilized by small molecule ligands, but it is unclear how the structure can be used for this. Has iduronate been modelled into the active site using the sulfate as anchor? This could help in structure-based design of weak inhibitors that could be displaced by substrate in the lysosome. Since any ligand binding anywhere on IDS will stabilize the enzyme, does the structure identify any potential binding sites/cavities away from the active site? Glycosylation can not only increase the solubility of proteins but also stabilize them from unfolding. Would the shielding oligosaccharides (an estimated 10-20 kDa from Suppl. Fig. 1a) interfere with a small molecule “chaperone” approach?

Minor:

- The sentence containing “co- or post-translationally oxidized” can be understood that the folded sulfatase precursor can be oxidized to FGly, which is not the case as the sulfatases need to be unfolded for modification (see for example the FGE/peptide complex structure where the peptide is linear as opposed to helical in the mature sulfatase). Sulfatases with a Cys do fold in vitro, are inactive, and can only partially be activated by harsh oxidation conditions. Please rephrase the sentence to include the “unfolded” aspect.

- “The expected severity of these mutations is variable and consequences range from local destabilization to global misfolding and premature degradation.” The term “misfolding”, here and at other places in the manuscript, implies an alternative native state, e.g. a filament as in Alzheimer’s, but the authors probably mean “unfolding”.
- “...is facilitated by deprotonation of the second germinal hydroxyl ...”. I like this typo.
- Last paragraph of the Discussion: “The structure of IDS described here provides a much-needed snapshot of the defective enzyme in Hunter syndrome”. I think you mean “...of the enzyme defective in Hunter syndrome.” since the wild-type IDS was crystallized.
- Table 1 reports no Ramachandran outliers, but I see 5 in Coot with the coordinates provided. Two of them are in loops of poor electron density. A genuine outlier is Lys347 in the active site, which is interesting because steric strain is often seen in enzyme active sites and stores energy for catalysis. It is useful to state which program was used for the Ramachandran statistics. Maybe other sulfatases can be checked for “outliers” in their active sites to see if this is a genuine feature.
- Table 1 should contain phasing information.
- Fig. 1b: Apart from the substituents, the glycosidic bonds are also different between heparin sulfate and dermatan sulfate. This could be added in the legend.
- Methods: Why is Suppl. Fig. 1a needed if the glycosylated material was used for crystallization? Since both, glycosylated and non-glycosylated IDS apparently crystallize, are the cell dimensions and space group for the non-glycosylated form known? If they are identical, the argument with the disordered oligosaccharides providing the crystal packing (see above) cannot be made.
- It is not shown if addition of 2.5 M urea is enough for denaturation of IDS. One would need a spectroscopic probe (CD or Trp fluorescence) to follow the unfolding. The PAGE in Suppl. Fig. 1a shows the increased suitability of IDS as a substrate for PNGase F as a function of [urea]. Urea might merely increase the flexibility of the protein (or its oligosaccharides), not unfold it.
- Methods: Why is Pb^{2+} needed for the enzymatic assay? The assay needs a reference or an explanation why Pb^{2+} , which is usually detrimental to protein health, is required, and also how it was ensured that the helper reactions in the coupled enzyme assay are not rate-limiting.
- Methods: “0.1 M MES pH 6.1” needs a corresponding base.
- Why was the iodide soak not collected at longer wavelengths? Was radiation damage an issue?
- In the coordinates and maps provided, there are several spurious water molecules with 2Fo-Fc density of <1 rmsd. One of them binds to the sulfate ion in the active site. I suggest removing them, as they could confuse users of the structure who do not look at maps or B-values.
- Suppl. Fig 1b: please state that the glycosylated IDS was used for the assay – it is stated in the main text.

Reviewer #2 (Remarks to the Author):

In this manuscript Demydchuk et al determine and describe the crystal structure of the recombinant human iduronate-2-sulfatase (IDS). The origin of a rare X-linked lysosomal storage disease known as Hunter syndrome is directly attributable to the loss of enzyme function due to mutations in the IDS gene. A large number of these mutations are known and the authors confirm by mapping them onto the structure that these mutations, indeed, impede the enzyme's catalytic activity and/or interfere with the folding. Hence, the structure is a welcome addition to the already known human and bacterial sulfatase family of enzymes. The manuscript is technically sound and well written. I only have a couple of minor technical points stated below.

However, despite having only about 20% sequence identity, IDS has essentially the same overall fold and tertiary structure as the known sulfatases. This is not at all surprising, as many families/superfamilies of proteins are known to possess identical folds even with 20-30% sequence conservation. Furthermore, the catalytic residues of IDS and their 3-D disposition around the active site are virtually identical to the known sulfatase structures, and so is the proposed mechanism of hydrolysis. Thus, in my estimation, the present work does not convey any new structural and/or biochemical data or insights on IDS or on the sulfatase family that could not have been derived simply by sequence alignment/homology modeling. What is missing in the field of sulfatases is the knowledge of substrate recognition mechanism. The structure of a complex of IDS with a non-hydrolysable substrate analog would have been a novel addition to the current understanding of the field.

I, therefore, do not recommend publication of the manuscript in Nature Communications on grounds of lack of novelty and insufficient conceptual advance.

Minor points:

1. The authors present enzyme activity (page 4, Results and supplementary Fig 1b) and structural data of the glycosylated IDS samples. It would be interesting to know if deglycosylated IDS is enzymatically active since the authors note that "Crystallisation trials were carried out with both glycosylated and deglycosylated protein, but as no improvement in diffraction quality was observed after deglycosylation, the glycosylated material was used for structure determination" (page 12, Methods). Are the crystals of glycosylated and deglycosylated IDS isomorphous? Is there any local structural change upon deglycosylation?
2. A typo in Table 1: Wavelength 0.9728Å or 0.9763Å? Also, data in the highest resolution shell appear to be rather weak ($I/\sigma I < 2.0$).

Reviewer #3 (Remarks to the Author):

The study of protein structure, in this case the sulfatases and its crystallization has important practical significance: for bioinformatics and prediction of disease severity and may make it easier to find the mutations amenable to a potential treatment with chaperons, in the future.

The work has valuable practical significance.

Sulphatase crystallization method is not novel but for idursulfatase important because MPS II is one of the most common MPS.

I wanted to point out the authors of the work Węgrzyn et al, 2010 Why are behaviors of children suffering from various neuronopathic types of mucopolysaccharidoses different? Węgrzyn G. et al, 2010 concerning the catabolism glikozaminoglikanów and the clinical / behavioral phenotype.

Reviewer #1

This manuscript describes the long-awaited crystal structure of IDS. The structure provides a platform to explain the phenotypes observed in many variants leading to Hunter syndrome, and may offer routes to the therapy of patients that have specific IDS variants with residual activity. I have these comments:

Major

- The packing shows that no crystal contacts exist along the a-axis, which is in principle physically impossible, although there are several legitimate examples in the PDB. As the glycosylated material was used for crystallization, disordered oligosaccharides could provide unspecific crystal contacts, but this is nowhere stated. Was there strong anisotropy in the data? The statistics in Table 1 appear very good for the resolution (although Rmerge is deprecated).*

The packing interactions are indeed so sparse that a local view around the unique molecule gives the impression that there are gaps in the packing. However, there are in fact crystal packing interactions along all three axes, as shown in the new Supplementary Fig. 2. The IDS molecules form filaments along the *c* (vertical) axis, which touch each other only twice per unit cell in each direction. As the referee correctly surmises, the crystals are significantly anisotropic, with substantially poorer diffraction in the *hk0* plane than along *00l*. The difference in the anisotropic B-factors between the weakest and strongest directions was about 70Å² for the first native data set and about 120Å² for the GdCl₃ data set. After testing a large number of crystals, the final native data set was fortunately less anisotropic, with a B-factor difference of only 38Å². We have added a line to table 1 giving the anisotropic deltaB values for all final data sets, and we mention the loose packing and consequent diffraction anisotropy in the main text.

- The sentence “An alternative interpretation that this is a phosphate picked up from the storage buffer...” invites mass spectrometry experiments, e.g. production of proteolytic fragments at pH where the esters are stable, followed by mass analysis.*

The alternative interpretation is mentioned as a formal possibility, because the enzyme is supplied as Elaprased with a high concentration of phosphate in the buffer. [Redacted]

[Redacted] However, as the referee notes in the next point, sulfate would be more stable and more likely to be encountered. We now state that the higher stability of sulfate esters makes this interpretation more likely.

- *The biochemical reason why sulfatases have FGly is that sulfate esters are more difficult to hydrolyze than are phosphate esters. The related serine phosphatases need only a Ser in their active site because the enzyme intermediate has higher energy. Sulfatases with a Ser instead of FGly are stuck at the stable sulfated enzyme intermediate. The enzyme-sulfate intermediate in active sulfatases is resolved by the elimination reaction. This circumstance would hint towards a sulfate, as suggested by the authors, rather than a phosphate in the IDS structure. The question is why a sulfate is trapped in the first place? This should be discussed by the authors. Possibly, the Lewis acidity of Ca²⁺ is insufficient? That calcium is present in the structure is convincingly shown, but where does it come from? As far as I know, the physiological ion is Mg²⁺, which is a stronger Lewis acid than Ca²⁺. And since Mg²⁺ is coordinated strictly octahedrally, the distorted configuration seen in the IDS structure might be an off-pathway structure that might have enabled trapping of the sulfate ester.*

We thank the referee for these helpful comments. As far as we can tell, there has been no systematic published investigation into the metal requirements for IDS, though the Shire literature on Elaprasedipine asserts that a calcium ion is required for activity. If the referee is aware of relevant literature on the physiological metal ion we would be grateful for a pointer. Nonetheless, both Mg²⁺ and Ca²⁺ have been identified as the active site metal ion in related phosphatase, sulfatase and sulfamidase enzymes (with Ca²⁺ being more common in crystal structures of the sulfatases), so we believe that either is possible *a priori*. Any metal in our protein preparation was carried through from the expression system, in which both magnesium and calcium would have been present, but no metal ions are added to the Elaprasedipine buffer or to our crystallization conditions. The implied tight binding for the metal ion and the clear evidence (bond lengths, coordination geometry, ligands) that what we see is Ca²⁺ suggests to us that this is probably the physiologically relevant metal.

The point about Lewis acidity is very pertinent. This explains a clear correlation among related structures catalyzing a variety of hydrolysis reactions: the majority of these do have Mg²⁺ modeled in the active site, but all 6 other related structures containing a covalently modified formylglycine (sulfate ester or phosphate ester) have Ca²⁺ modeled in the active site (and with coordination geometry consistent with that interpretation). We now comment on these points in the revised manuscript: “With its lower Lewis acidity, Ca²⁺ is more consistent than Mg²⁺ with the stability of the observed sulfate ester modification of formylglycine: in related structures, the identification of either a sulfate ester (ARSB, STS) or phosphate ester (ARSA, SGSH) modification correlates with the identification of the metal ion as Ca²⁺.”

- *Regarding the mechanism in Fig. 3e: I have yet to find a single structure of a carboxylic acid or an alcohol bound to a metal ion in the CSD that has convincing electron density for the proton(s), but maybe the authors know of examples? It appears that if a metal ion is bound to oxygen, the proton generally dissociates, even if this means a surplus of negative charge next to the metal ion that has to be neutralized elsewhere. Hence, the top left part of the figure where FGH binds Ca²⁺ and retains its proton might be wrong. FGH might already be present as an anion, stabilized by the positive potential in the active site. While oxyanions in phosphatases and proteases have to be stabilized by many interactions (NH-groups in the oxyanion hole”, the helix dipole), the additional hydroxyl group of hydrated FGly reduces the pKa of either hydroxyl, and metal ion binding might lead to deprotonation, so the nucleophilic anion could be a stable ground state. The positive end of the helix dipole where FGly is located will also contribute to stabilization of an anion.*
- *“...and by D334 donating a proton to the departing sulfate”. This is shown in the bottom right panel of Fig. 3e. Since HSO₄⁻ has a pKa of 2, it is unlikely for Asp334 to donate a proton. It is also not needed, since sulfate is a very good leaving group. I would argue that Asp334, being bound to the metal ion, is deprotonated throughout the catalytic cycle and does not need to play an active role.*
- *“The final rehydration of the resultant FGly aldehyde is assisted by activation of water by H229 and*

protonation by H138". With "protonation" you probably mean "protonation of the hydrated aldehyde", not the water molecule. H229 is solvent-exposed, and thus very likely always protonated at ambient pH (lysosome) unless shielded by the substrate (a model could help here, see also below). Alternatively, water may attack FGly and the proton is then removed by other waters. The Grothaus mechanism would be a feasible route for the proton to exit the active site. Since the insights into the sulfatase mechanism provided by this structure are limited, I suggest keeping its discussion at a minimum.

We thank the referee for these interesting mechanistic suggestions, and we agree that whilst the mechanism presented in Fig 3e is based on the consensus TE mechanism for similar sulfatases, the exact mechanistic details are not a direct result of our work. We have rewritten this part of the discussion to avoid assigning definite roles to individual residues where there is still ambiguity in the field concerning e.g. proton balance. We have also replaced panel e in Fig. 3 with docking experiments (see below).

- *Despite the presence of the sulfate, IDS was active at low pH. This could be of biological significance – inactive enzyme during transport, activation at its target location. However, the notion invites more experiments: A new structure after pre-treatment of IDS at lower pH would tell whether this hypothesis is true (the sulfate should be absent). Addition of Mg²⁺ to the crystallization experiment could replace the Ca²⁺. Alternatively, MS analysis at different pH could help (see second point above).*

We did attempt to treat the enzyme in various ways to remove the covalently attached sulfate group. Unfortunately, we did not succeed in crystallising the enzyme after any of these treatments.

- *The Discussion makes a strong point of the possibility that certain Hunter syndrome-causing variants of IDS can be stabilized by small molecule ligands, but it is unclear how the structure can be used for this. Has iduronate been modelled into the active site using the sulfate as anchor? This could help in structure-based design of weak inhibitors that could be displaced by substrate in the lysosome.*

This is an excellent suggestion and we have now performed docking experiments with 2-sulfo-iduronic acid, heparan sulfate disaccharides and a dermatan sulfate disaccharide to gain some insight into possible modes of substrate binding and the involvement of other residues. An FGH residue was modelled in the active site instead of FGS, and the position of the scissile sulfate group was restrained appropriately during docking (based on the high-resolution structure of non-covalently bound sulfate in PAS, see supplementary Fig 3b). We have added a methods section to describe this, and the results are presented in a new version of Fig 3. We have also included a B-factor analysis of all residues surrounding the active site to indicate which parts of the binding site are very well-ordered and likely to be pre-organized versus residues that are likely more flexible. This is now discussed in the text.

- *Since any ligand binding anywhere on IDS will stabilize the enzyme, does the structure identify any potential binding sites/cavities away from the active site? Glycosylation can not only increase the solubility of proteins but also stabilize them from unfolding. Would the shielding oligosaccharides (an estimated 10-20 kDa from Suppl. Fig. 1a) interfere with a small molecule "chaperone" approach?*

The oligosaccharides are all sufficiently far from the active site that they should not interfere with binding of any active-site directed pharmacological chaperones, and we have added an additional supplementary figure (Fig. S5) to illustrate the presence of several other large cavities on the IDS electrostatic surface that could potentially be exploited for pharmacological chaperone development.

Minor:

- *The sentence containing "co- or post-translationally oxidized" can be understood that the folded sulfatase precursor can be oxidized to FGly, which is not the case as the sulfatases need to be unfolded for modification (see for example the FGE/peptide complex structure where the peptide is linear as opposed to helical in the mature sulfatase). Sulfatases with a Cys do fold in vitro, are inactive, and can only partially be activated by harsh oxidation conditions. Please rephrase the sentence to include the*

“unfolded” aspect.

We agree that the idea of post-translational oxidation is incompatible with the modification taking place during (co-translational) translocation into the ER and have therefore removed the misleading words “co- or post-translationally oxidized”. We have also changed “nascent polypeptide” to “nascent (and unfolded) polypeptide” in the subsequent sentence.

- *“The expected severity of these mutations is variable and consequences range from local destabilization to global misfolding and premature degradation.” The term “misfolding”, here and at other places in the manuscript, implies an alternative native state, e.g. a filament as in Alzheimer’s, but the authors probably mean “unfolding”.*

We feel that the word “misfolding”, commonly used in this area, captures the idea of what happens when only part of the protein structure is destabilized, but the phrase “global misfolding” could be misleading. We have rephrased the relevant part as “local destabilization and misfolding to global unfolding, leading to premature degradation”.

- *“...is facilitated by deprotonation of the second germinal hydroxyl ...”. I like this typo.*

Fixed, thanks.

- *Last paragraph of the Discussion: “The structure of IDS described here provides a much-needed snapshot of the defective enzyme in Hunter syndrome”. I think you mean “...of the enzyme defective in Hunter syndrome.” since the wild-type IDS was crystallized.*

Fixed, thanks.

- *Table 1 reports no Ramachandran outliers, but I see 5 in Coot with the coordinates provided. Two of them are in loops of poor electron density. A genuine outlier is Lys347 in the active site, which is interesting because steric strain is often seen in enzyme active sites and stores energy for catalysis. It is useful to state which program was used for the Ramachandran statistics. Maybe other sulfatases can be checked for “outliers” in their active sites to see if this is a genuine feature.*

The few outliers flagged by Coot are in regions of Ramachandran space that the current version of Molprobity (based on a larger database of high quality structures) interprets as relatively rare but allowed. We agree that the source of the validation criteria should be defined and now state that Molprobity was used for the Ramachandran statistics.

- *Table 1 should contain phasing information.*

We would agree with this in principle, but the process of phasing was sufficiently unconventional that the normal phasing statistics are rather difficult to interpret. The process is described in detail in Methods, and we feel this is more informative than the conventional summary statistics.

- *Fig. 1b: Apart from the substituents, the glycosidic bonds are also different between heparin sulfate and dermatan sulfate. This could be added in the legend.*

The linkage is now specified in the legend.

- *Methods: Why is Suppl. Fig. 1a needed if the glycosylated material was used for crystallization? Since both, glycosylated and non-glycosylated IDS apparently crystallize, are the cell dimensions and space group for the non-glycosylated form known? If they are identical, the argument with the disordered oligosaccharides providing the crystal packing (see above) cannot be made.*

This figure is provided in connection with estimating the mass of the protein without glycans. The deglycosylated protein crystallized with cell dimensions within the range seen for glycosylated protein, the crystals of which tend to be poorly isomorphous. As discussed above, the crystal packing contacts can be seen without having to invoke disordered oligosaccharides. We now mention in the text that the crystal forms were isomorphous.

- *It is not shown if addition of 2.5 M urea is enough for denaturation of IDS. One would need a spectroscopic probe (CD or Trp fluorescence) to follow the unfolding. The PAGE in Suppl. Fig. 1a*

shows the increased suitability of IDS as a substrate for PNGase F as a function of [urea]. Urea might merely increase the flexibility of the protein (or its oligosaccharides), not unfold it.

It is true that the protein does not necessarily have to unfold completely to allow deglycosylation, and this experiment only shows that about 1.0M urea is sufficient to achieve complete deglycosylation. So we have changed “denaturation” to “partial denaturation” in the legend.

- *Methods: Why is Pb²⁺ needed for the enzymatic assay? The assay needs a reference or an explanation why Pb²⁺, which is usually detrimental to protein health, is required, and also how it was ensured that the helper reactions in the coupled enzyme assay are not rate-limiting.*

The references underlying the Moscerdam Substrates assay are already cited: Voznyi *et al.* and Tolun *et al.* In the underlying research, they showed that the assay is linear with enzyme concentration. They suggest that Pb²⁺ acts by precipitating sulfates and phosphates, which could otherwise inhibit the enzyme activity. Indeed, we were surprised by the requirement for this ion, but a control experiment demonstrated that it was also required in our hands.

- *Methods: “0.1 M MES pH 6.1” needs a corresponding base.*

Thanks, fixed (MES-NaOH).

- *Why was the iodide soak not collected at longer wavelengths? Was radiation damage an issue?*

The first attempt to collect iodide soak data was made at the iodide peak wavelength, but radiation damage was indeed an issue that prevented us from using those data. We have added a note to this effect in the text.

- *In the coordinates and maps provided, there are several spurious water molecules with 2Fo-Fc density of <1 rmsd. One of them binds to the sulfate ion in the active site. I suggest removing them, as they could confuse users of the structure who do not look at maps or B-values.*

The refinement of a large structure is never absolutely complete, and there are bound still to be debatable minor features even if we add to the many rounds of rebuilding and refinement already carried out. Any waters in the model would have been placed to explain significant positive density features, so leaving them out is also likely to make the refinement look incomplete.

- *Suppl. Fig 1b: please state that the glycosylated IDS was used for the assay – it is stated in the main text.*

Done.

We greatly appreciate the detailed suggestions for improvement made by this reviewer. As he signed his review, we have now acknowledged his help in the revised manuscript.

Reviewer #2

In this manuscript Demydchuk et al determine and describe the crystal structure of the recombinant human iduronate-2-sulfatase (IDS). The origin of a rare X-linked lysosomal storage disease known as Hunter syndrome is directly attributable to the loss of enzyme function due to mutations in the IDS gene. A large number of these mutations are known and the authors confirm by mapping them onto the structure that these mutations, indeed, impede the enzyme's catalytic activity and/or interfere with the folding. Hence, the structure is a welcome addition to the already known human and bacterial sulfatase family of enzymes. The manuscript is technically sound and well written. I only have a couple of minor technical points stated below.

However, despite having only about 20% sequence identity, IDS has essentially the same overall fold and tertiary structure as the known sulfatases. This is not at all surprising, as many families/superfamilies of proteins are known to possess identical folds even with 20-30% sequence conservation. Furthermore, the catalytic residues of IDS and their 3-D disposition around the active site are virtually identical to the known sulfatase structures, and so is the proposed mechanism of hydrolysis. Thus, in my estimation, the present work does not convey any new structural and/or biochemical data or insights on IDS or on the sulfatase family that could not have been derived simply by sequence alignment/homology modeling. What is missing in the field of

sulfatases is the knowledge of substrate recognition mechanism. The structure of a complex of IDS with a non-hydrolysable substrate analog would have been a novel addition to the current understanding of the field.

I, therefore, do not recommend publication of the manuscript in Nature Communications on grounds of lack of novelty and insufficient conceptual advance.

We agree completely with the referee that the structure of a complex with a substrate analog is very desirable, which is why we delayed publication for a long time while attempting to bind a wide variety of potential ligands. Unfortunately, these attempts did not succeed. We have now performed docking experiments to generate insight into possible modes of substrate binding (see related point by referee 1, above).

However, we disagree with the referee on the degree to which insights could have been derived by sequence alignment or homology modeling. At the time of the structure determination, the closest relatives had sequence identities in the low 20% range, and there are still no relatives of known structure with as high as 30% identity over a substantial part of the molecule. At this limited level of sequence identity, the core fold and the key catalytic residues are indeed preserved, but there is great structural variation in the regions outside the core and there is great uncertainty even in the sequence alignments. Given that any mutations that disrupt the fold – not just in the core or active site – will cause disease, the actual structure is essential for a proper understanding of disease-associated mutations. As an indication of the uncertainties associated with modeling from relatively distant homologues, I have retrieved the ModBase model of IDS, which was derived last year using PDB entry 4upi as a template. The figure below shows that, even though the core is basically correct, large parts of the structure are completely incorrect or unmodeled. The model covers only 402 residues of the 517 in mature IDS, only 301 residues of the model superimpose with an RMSD of 2.0Å, and of these only 244 (*i.e.* less than half of the complete structure) have the correct sequence register.

Minor points:

1. The authors present enzyme activity (page 4, Results and supplementary Fig 1b) and structural data of the glycosylated IDS samples. It would be interesting to know if deglycosylated IDS is enzymatically active since the authors note that “Crystallisation trials were carried out with both glycosylated and deglycosylated protein, but

as no improvement in diffraction quality was observed after deglycosylation, the glycosylated material was used for structure determination” (page 12, Methods). Are the crystals of glycosylated and deglycosylated IDS isomorphous? Is there any local structural change upon deglycosylation?

We did not pursue enzyme activity experiments with the deglycosylated material, as our main concern was to establish that the material in the crystal was active. As noted above, we now mention that the two crystal forms are isomorphous. After screening a large number of crystals to find the one that diffracted most isotropically and to the highest resolution, we did not go back to the earlier (and poorer) crystals from deglycosylated material to see if there were any structural changes, as this did not seem relevant to our goals of understanding the disease-associated mutations.

2. A typo in Table 1: Wavelength 0.9728Å or 0.9763Å? Also, data in the highest resolution shell appear to be rather weak (<2.0).

We thank the referee for catching this typo: both the native and NaI data were collected with a wavelength of 0.9763Å. However, we do not agree that the data in the highest resolution shell are too weak. In recent years, particularly through the work of Karplus & Diederichs, it has been learned that modern refinement programs can extract information from weaker data than previously appreciated. In fact, our resolution cutoff is on the stringent side by the criteria of Karplus & Diederichs, in which one would usually allow $CC_{1/2}$ to drop to 0.5 or even somewhat lower. We found, through trial refinements, that using data to higher resolution degraded the quality of the results. We suspect that impending improvements in the treatment of measurement error in refinement likelihood scores (work that we are involved in) will further change the picture, and we look forward to returning to these data at a later date to see if we can add back some of the higher resolution data.

Reviewer #3

The study of protein structure, in this case the sulfatases and its crystallization has important practical significance: for bioinformatics and prediction of disease severity and may make it easier to find the mutations amenable to a potential treatment with chaperons, in the future.

The work has valuable practical significance.

Sulphatase crystallization method is not novel but for idursulfatase important because MPS II is one of the most common MPS.

I wanted to point out the authors of the work Węgrzyn et al, 2010 Why are behaviors of children suffering from various neuronopathic types of mucopolysaccharidoses different? Węgrzyn G. et al, 2010 concerning the catabolism glikozaminoglikanów and the clinical / behavioral phenotype.

We thank the referee for bringing this work to our attention. Indeed, the correlation between genotype and phenotype is a fascinating question, and the actual disease mechanisms for many diseases are only poorly understood if at all. We now cite an earlier paper from the same group (Piotrowska *et al.*, 2009), showing that the prediction of phenotype can be strengthened by adding knowledge of the patient's level of GAG synthesis. Our structure reveals the molecular details of the defects leading to the accumulation of GAGs, but understanding other aspects of the upstream and downstream processes leading to disease will also be essential.

REVIEWERS' COMMENTS:

Reviewer #1 (Remarks to the Author):

The manuscript looks fine. Thank you for including me in the Acknowledgement section, I didn't expect this.

M. Rudolph